# The nuclear charge radius of $^{13}$C

Patrick Müller [1,5] ✉, Matthias Heinz [1,2,3,6], Phillip Imgram [1,7], Kristian König [1,4], Bernhard Maass [1,8], Takayuki Miyagi [1,2,3,9], Wilfried Nörtershäuser [1,4] ✉, Robert Roth [1,4] & Achim Schwenk [1,2,3]

The size is a key property of a nucleus. Accurate nuclear radii are extracted from elastic electron scattering, laser spectroscopy, and muonic atom spectroscopy. The results are not always compatible, as the proton-radius puzzle has shown most dramatically. Beyond helium, precision data from muonic and electronic sources are scarce in the light-mass region. The stable isotopes of carbon are an exception. We present a laser spectroscopic measurement of the root-mean-square (rms) charge radius of $^{13}$C and compare this with ab initio nuclear structure calculations. Measuring all hyperfine components of the $2\,^3S \rightarrow 2\,^3P$ fine-structure triplet in $^{13}C^{4+}$ ions referenced to a frequency comb allows us to determine its center-of-gravity with accuracy better than 2 MHz although second-order hyperfine-structure effects shift individual lines by several GHz. We improved the uncertainty of $R_c(^{13}C)$ determined with electrons by a factor of 6 and found a $3\sigma$ discrepancy with the muonic atom result of similar accuracy.

The main sources for absolute nuclear charge radii of stable isotopes are elastic electron scattering[1] and muonic atom spectroscopy[2,3], i.e., the energy determination of X-rays emitted in $K_\alpha$ transitions in muonic atoms. The combination of both is considered to deliver the most accurate values, which serve as anchor points for the determination of charge radii along an isotopic chain using isotope-shift measurements in rare and short-lived species[4,5]. For the proton's size, electronic measurements, i.e., laser spectroscopy of hydrogen[6] and elastic electron scattering[7] on one side, and laser spectroscopy of muonic hydrogen on the other side[8], led to a surprising $7\sigma$ discrepancy of the proton's charge radius, which became famously known as the "proton radius puzzle". Even though laser and microwave spectroscopy of ordinary hydrogen become increasingly consistent with the smaller radius from muonic hydrogen[9–11], there are still measurements reporting larger radii[12,13]. Similarly, the source of the discrepancy in electron scattering is still under debate[14–16] and is a motivation for the MUSE experiment at PSI[17] with the goal to perform muon-proton scattering to test lepton universality in the electromagnetic

interaction. Contrary, in $^{3,4}$He, electron scattering and muonic atom laser spectroscopy are in good agreement[18,19], while the resulting difference in mean-square (ms) charge radius is in tension with several results from laser spectroscopy of ordinary helium[20]. These are, however, affected by a long-standing disagreement—depending on the transition and the applied technique, they scatter by several $\sigma$[21]. Very recently, these tensions in charge radii differences of $^3$He and $^4$He are partially resolved by taking into account second-order hyperfine corrections[22,23]. For the next elements, Li, Be, and B, nuclear charge radii are about two orders of magnitude less precise than in hydrogen or He and for N, O, and F, it is only slightly better[5,24]. Second-generation muonic X-ray experiments are in preparation that will apply magnetic calorimeters and use improved theoretical calculations to extract absolute nuclear radii of stable isotopes from beryllium to neon[24].

The only exception in the second row of the periodic table is carbon. The charge radius of $^{12}$C was determined with exceptional 2 and 1 per mille accuracy by electron scattering[25] and muonic atom X-ray spectroscopy[26], respectively, but a small $2\sigma$ discrepancy between

[1]Institut für Kernphysik, Technische Universität Darmstadt, Darmstadt, Germany. [2]ExtreMe Matter Institute EMMI, GSI Helmholtzzentrum für Schwerionenforschung GmbH, Darmstadt, Germany. [3]Max-Planck-Institut für Kernphysik, Heidelberg, Germany. [4]Helmholtz Research Academy Hesse for FAIR, GSI Helmholtzzentrum für Schwerionenforschung, Darmstadt, Germany. [5]Present address: Department of Physics and Astronomy, University of California Los Angeles, Los Angeles, CA, USA. [6]Present address: National Center for Computational Sciences, Oak Ridge National Laboratory, Oak Ridge, TN, USA. [7]Present address: Instituut voor Kern- en Stralingsfysica, KU Leuven, Leuven, Belgium. [8]Present address: Physics Division, Argonne National Laboratory, Argonne, IL, USA. [9]Present address: Center for Computational Sciences, University of Tsukuba, Tsukuba, Japan. ✉e-mail: pmueller@physics.ucla.edu; wnoertershaeuser@ikp.tu-darmstadt.de

the two values is observed. For $^{13}$C, the second stable isotope, results from both methods agree within the comparably large uncertainty of the electron-scattering result. Thus, there is a demand for more precise knowledge of the charge radius of $^{13}$C from the electronic sector. Moreover, radii of light nuclei are also particularly interesting for nuclear structure theory as they are accessible to different ab initio methods and show substantial structural changes between neighboring isotopes[27]. A reason for this is the tendency of the nucleons to form $\alpha$ clusters, which in light nuclei constitute a large portion of the nucleus[28–35]. To reproduce the size of the firmly bound $p_{3/2}$ subshell-closed nucleus of $^{12}$C with ab initio nuclear structure calculations, a significant admixture of a $3\alpha$-cluster configuration is required that is governed by the $0_2^+$ Hoyle state[36–40]. This characteristic structure is essential in the $3\alpha$-nucleosynthesis process in stars and supernovae[41–43]. In $^{13}$C, the additional neutron may be thought to act as a covalent bond inside the $3\alpha$ structure[32], explaining the smaller radius of $^{13}$C compared with $^{12}$C as it was determined consistently by elastic electron scattering and muonic atom spectroscopy. The experimental accuracy of the $^{13}$C radius solely based on electronic measurements can be improved by combining the accurate $^{12}$C electron scattering result with a precise determination of the differential ms charge radius $\delta\langle r^2\rangle^{12,13}$ extracted from an isotope-shift measurement.

So far, there is no laser-spectroscopic nuclear structure information for the elements beyond Be[44,45] up to neon[46]. This holds for the stable and short-lived isotopes except for a measurement of the isotope shift in stable $^{10,11}$B[35]. Efforts in the laser-spectroscopic determination of charge radii in this region have to face three main challenges: Firstly, the electronic level schemes of the elements above boron are complicated, and excitation from the ground state of atoms or singly charged ions is not feasible with currently available high-precision lasers. Secondly, high-accuracy mass-shift calculations for the systems beyond boron, which are required to extract the nuclear charge radius, are unavailable. Finally, due to their chemical reactivity, most of these elements can only be extracted in the form of molecules at online isotope separator facilities, while at in-flight facilities, the stopping of such light products requires large stopping cells and the yields are comparably low. Here, we demonstrate an approach that opens up prospects for such measurements by simultaneously overcoming all three hurdles.

Our experimental method is based on generating He-like ions−in this case, C$^{4+}$ ions−in the excited metastable $1s2s\,^3S_1$ state. From here, laser excitation into the $1s2p\,^3P_J$ manifold is possible with ultraviolet light. This transition has been previously studied in $^{12}$C$^{4+}$ to perform a test of non-relativistic quantum electrodynamics (NRQED) calculations and to demonstrate that an "all-optical" nuclear charge radius extraction with high accuracy is potentially possible[47,48]. While the charge radius of $^{12}$C can, in principle, be extracted directly from the transition frequency, theory is not yet sufficiently advanced to make this "all-optical" approach competitive with the alternative methods[47]. In this work, we present a measurement of the charge radius of $^{13}$C based on an optical isotope-shift measurement and NRQED mass-shift calculations with accuracy that already exceeds the available electron-scattering data on $^{13}$C. Since the isotope shift, i.e., the difference in transition frequency between two isotopes with mass numbers $A$ and $A'$

$$\delta\nu^{A,A'} = \nu^{A'} - \nu^A = \delta\nu_M^{A,A'} + F\,\delta\langle r^2\rangle^{A,A'}, \qquad (1)$$

it thus follows that the change in the ms nuclear charge radius can be extracted according to

$$\delta\langle r^2\rangle^{A,A'} = \frac{\delta\nu^{A,A'} - \delta\nu_M^{A,A'}}{F} \qquad (2)$$

in a nuclear-model independent way, provided that the mass-shift $\delta\nu_M$ and the field-shift factor $F$ can be reliably calculated.

This approach is currently only possible for up to five-electron systems[35]. Hence, spectroscopy on neutral carbon atoms is not an option, but mass-shift calculations for the He-like system C$^{4+}$ are readily available, and the field-shift factor of −211.5(1) MHz/fm$^2$ in the laser-accessible $1s2s\,^3S_1 \rightarrow 1s2p\,^3P_{0,1,2}$ transition provides a high sensitivity to the charge radius. Our measurement of the charge radius of $^{13}$C makes the $^{12,13}$C pair the nuclei with the currently best-known nuclear charge radii besides the stable isotopes of hydrogen and helium, for which laser spectroscopy was performed on muonic atoms[18,19,49]. We finally note that this has been achieved despite strong perturbations by hyperfine-induced fine-structure mixing and required high-precision measurements of all nine hyperfine components in the three fine-structure lines.

## Results

We performed the measurements at the COllinear Apparatus for Laser Spectroscopy and Applied Sciences (COALA) at the Institute for Nuclear Physics at TU Darmstadt. The setup and the recent extensions for highly charged ions are described in refs. 48,50, and we only briefly summarize the most relevant aspects for our work. A sketch of COALA and the measurement principle is depicted in Fig. 1. A continuous beam of $^{13}$C$^{4+}$ ions with a beam current of approximately 1.5 nA and an energy of 50 keV is produced in our electron-beam ion source (EBIS-A, DREEBIT GmbH). We feed $^{13}$C-enriched methane gas ($^{13}$CH$_4$) at a pressure of $6\cdot10^{-8}$ mbar into the EBIS. The molecules are cracked and charge states up to fully stripped C$^{6+}$ ions are reached through subsequent collisions of the ions with the electron beam. The ions are confined by the electrostatic potentials of the trap electrodes and the electron beam's space charge in axial and radial directions, respectively. The starting point for laser spectroscopy is the metastable $1s2s\,^3S_1$ state in He-like $^{13}$C$^{4+}$ with a lifetime of 21 ms[51]. It is dominantly populated through electron-capture processes of C$^{5+}$[48]. The electrode generating the axial trapping potential at the trap exit is only 200 V above the central trap potential ($\approx$+12.5 kV). Thus, a small fraction of the trapped ions continuously leaks out of the trap and is accelerated toward ground potential. A velocity filter (Wien filter) formed by crossed magnetic and electric fields is used to separate the $^{13}$C$^{4+}$ ions from other charge states and ion species. Behind the source region, an electrostatic 60°-bender, followed by xy-steerers and a quadrupole doublet, is used to superimpose the ion-beam with the laser beams and to shape the ion beam, respectively. The fluorescence detection region (FDR) is floated on a scan voltage $U_{scan}$ to adjust the ion velocity and thereby scan the laser frequency in the rest-frame of the ions (Doppler tuning).

Two continuous-wave Ti:sapphire lasers produce light between 906 and 914 nm that is twice frequency doubled to create the range from 226.5 to 228.5 nm required to perform collinear and anticollinear laser spectroscopy on the $^{13}$C$^{4+}$ ions in fast alternation. We note that the lines are shifted by ±0.6 nm due to the Doppler shift. The exact frequency depends on the ion velocity, but this dependency is eliminated if the resonance frequencies in both collinear $\nu_c$ and anticollinear $\nu_a$ direction are determined. The geometric average $\nu_0 = \sqrt{\nu_a \nu_c}$ of the observed resonance frequencies in the laboratory frame is the transition frequency in the rest frame of the ion. Both Ti:sapphire lasers are simultaneously locked to a frequency comb to obtain $\nu_a$ and $\nu_c$ with high accuracy.

A schematic of the hyperfine structure in the $1s2s\,^3S_1 \rightarrow 1s2p\,^3P_{0,1,2}$ multiplet is shown in Fig. 2a, and the observed spectrum is depicted in (b). The HFS is well resolved in all transitions and spans 60 to 90 GHz. The relative peak heights vary based on their dipole transition strengths, with the peak of the weakest transition being ten times smaller than that of the strongest. Due to the large hyperfine splitting and the narrow full width at half maximum of 150 MHz, the individual transitions are compressed to a line in the overview plot. Therefore, the fits are shown in the inset for the smallest (left) and the strongest (right) hyperfine components to indicate the typical statistical significance. Resonances are fitted with a pure Gaussian lineshape since

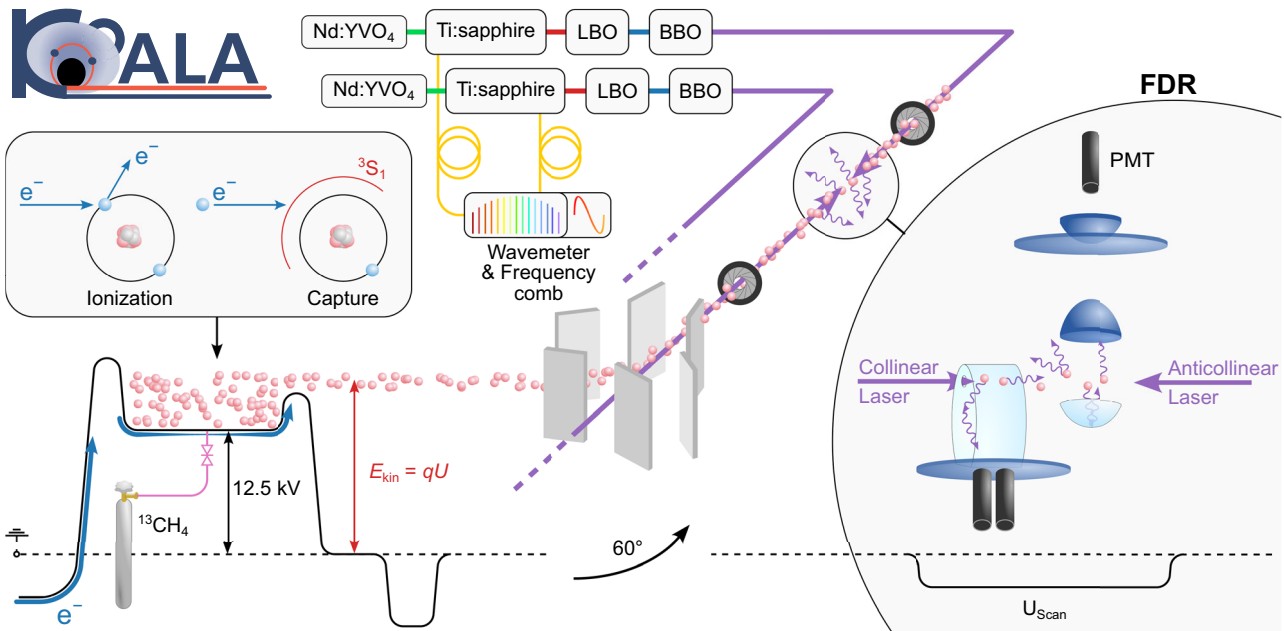

**Fig. 1 | Experimental setup.** Sketch of the measurement principle including the electron-beam ion source (EBIS), the electrostatic switchyard, the beam alignment irises, the laser system, and the fluorescence detection region (FDR). The laser system consists of two Millennia pump lasers (Nd:YVO₄) that drive two Ti:Sapphire lasers, each followed by two frequency doublers, one operated with lithium-

triborate (LBO) and second one with barium betaborate (BBO). The lasers are locked to a wavemeter and a frequency comb. The 227-nm light is then transported through air to the COALA beamline. The two charge-breeding processes, electron-impact ionization and electron capture, are shown in the inset above the EBIS potential.

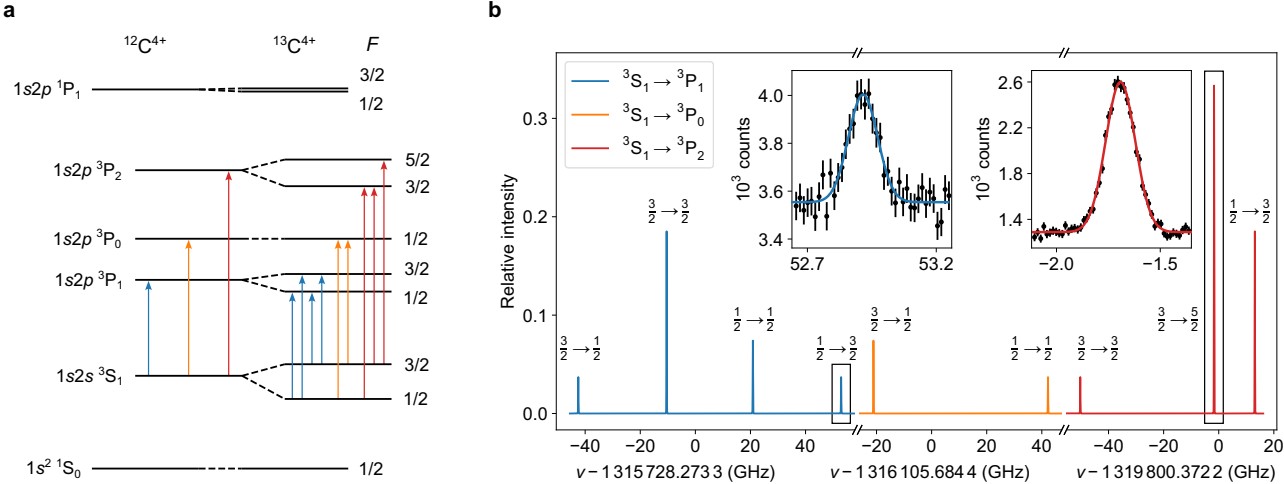

**Fig. 2 | Atomic spectroscopy data of $^{12,13}$C$^{4+}$. a** Level scheme of $^{12,13}$C$^{4+}$ and the electric dipole transitions that are addressed by the laser. **b** Hyperfine-structure (HFS) spectrum of the $1s2s\,^3S_1 \rightarrow 1s2p\,^3P_{0,1,2}$ transitions in $^{13}$C$^{4+}$ simulated with the experimentally determined frequencies and linewidths. The x-axis represents the laser frequency in the rest-frame of the ion $v$ relative to the center-of-gravity

frequency of the respective fine-structure transition. The peak heights were set to the theoretical transition strengths used in Eq. (7). The two insets show measured spectra of the marked transitions. Next to the resonances, the contributing quantum numbers $F \rightarrow F'$ of the lower and upper states are shown, respectively.

the natural linewidth of 9 MHz[51] and possible homogeneous broadening mechanisms are negligible compared to the width of the velocity distribution of the ions. The total statistical uncertainty of the resonance center is in all cases ≲ 1 MHz and therefore significantly smaller than potential systematic shifts.

The largest systematic uncertainty originates from the imperfect alignment of the two laser beams. If the collinear and the anticollinear laser beams are slightly offset in the detection region, they address different ion velocities and the Doppler shift is not fully canceled in the geometric average. Experimental verification of this effect provided a conservative limit of 1.7 MHz under the given experimental conditions[48]. This is the dominant contribution to the systematic

uncertainty. Adding all systematics in square (beam alignment, Zeeman effect, uncorrected photon recoils) yields a total systematic uncertainty of 1.8 MHz. Due to the linear dependence of the frequency shift on the statistical misalignment of the laser beams and the daily realignment of the ion and laser beams, we expect the dominant systematic uncertainty to fluctuate centered around the atomic transition frequency. Thus, the statistical and systematic uncertainties are added in square to obtain the total uncertainties. For details, see Methods.

The transition frequencies of all hyperfine-structure lines are compiled in Table 1. The individual transition frequencies of $^{12}$C$^{4+}$ from ref. 47 and the center-of-gravity (cg) frequency of the fine-structure multiplet are also included. For each fine-structure component of $^{13}$C,

**Table 1 | Electronic transition frequencies $\nu^{13}$ and isotope shifts $\delta\nu^{12,13}$ of the $1s2s\,^3S_{(J,F)}, F \to 1s2p\,^3P_{(J',F')}$ transitions in $^{13}C^{4+}$**

| $(J, F) \to (J', F')$ | $\nu^{13}_{(J,F)\to(J',F')}$ | $\delta\nu^{12,13}_{J\to J'}$ |
|---|---|---|
| $(1, 1/2) \to (0, 1/2)$ | 1 316 147 920.6 (1.9) | — |
| $(1, 3/2) \to (0, 1/2)$ | 1 316 084 566.3 (1.8) | — |
| $1 \to \bar{0}$ | 1 316 105 684.4 (1.4) | 53 465.1 (2.3) |
| $1 \to 0$ | 1 316 103 946.9 (1.3) | 51 727.6 (2.3) |
| $1 \to 0, {}^{12}C^{4+\ 47}$ | 1 316 052 219.3 (1.9) | — |
| $(1, 1/2) \to (1, 1/2)$ | 1 315 749 143.7 (1.9) | — |
| $(1, 1/2) \to (1, 3/2)$ | 1 315 781 189.1 (2.0) | — |
| $(1, 3/2) \to (1, 1/2)$ | 1 315 685 791.2 (2.0) | — |
| $(1, 3/2) \to (1, 3/2)$ | 1 315 717 838.5 (1.8) | — |
| $1 \to \bar{1}$ | 1 315 728 273.3 (1.1) | 51 080.5 (2.0) |
| $1 \to 1$ | 1 315 728 925.4 (1.0) | 51 732.6 (2.0) |
| $1 \to 1, {}^{12}C^{4+\ 47}$ | 1 315 677 192.8 (1.7) | — |
| $(1, 1/2) \to (2, 3/2)$ | 1 319 813 468.4 (1.8) | — |
| $(1, 3/2) \to (2, 3/2)$ | 1 319 750 116.8 (1.8) | — |
| $(1, 3/2) \to (2, 5/2)$ | 1 319 798 680.5 (1.8) | — |
| $1 \to \bar{2}$ | 1 319 800 372.2 (1.2) | 51 800.8 (2.1) |
| $1 \to 2$ | 1 319 800 329.2 (1.2) | 51 757.8 (2.1) |
| $1 \to 2, {}^{12}C^{4+\ 47}$ | 1 319 748 571.4 (1.7) | — |
| $S \to \bar{P}$ | 1 318 032 485.0 (0.8) | 51 745.6 (1.4) |
| $S \to P$ | 1 318 032 485.5 (0.8) | 51 746.1 (1.4) |
| $S \to P, {}^{12}C^{4+}$ | 1 317 980 739.4 (1.1) | — |

The center-of-gravity (cg) frequencies below the individual resonance frequencies are calculated using Eqs. (7) and (8), with and without hyperfine-induced mixing considered, respectively. The isotope shifts in the third column are the differences between the absolute transition frequencies in this table and the results in $^{12}C^{4+\ 47}$. All values are given in MHz.

the hyperfine cg was first obtained assuming the usual first-order hyperfine structure splitting (for details, see Methods) and is listed in the table indicated as $1 \to \bar{J}$. A look at the individual isotope shifts relative to the corresponding fine-structure transition in $^{12}C$ shows that these vary over 2.4 GHz even though they are expected to be equal. Second-order effects cause this discrepancy, i.e., the mixing of hyperfine states with the same quantum number $F$ belonging to different fine-structure levels. Including second-order HFS explicitly, using the theoretical magnetic dipole matrix elements tabulated in ref. 52 (see Methods), we obtain the cg as provided in the table indicated by $1 \to J$. In this case, the isotope shift of all three fine-structure components agree to $\lesssim 30$ MHz, demonstrating that the calculated magnetic properties capture the shifts induced by state mixing well. We attribute the remaining difference to the "splitting isotope shift" that provides an important benchmark for NRQED calculations[53]. The comparison between the values obtained with and without hyperfine mixing reveals that the shifts of the $^3P_{0,1}$ levels are much larger than the shift of the more separated $^3P_2$ level.

The isotope shifts of the $^3S_1 \to {}^3P_J$ fine-structure centroid determined by the two approaches differ by only 500 kHz. This difference is ascribed to the small contribution of the well-separated $^1P_1$ level. The excellent agreement, much smaller than our uncertainty, demonstrates that measuring all hyperfine components is a reliable way to circumvent the impact of hyperfine mixing and to extract the charge radius with high accuracy, which is particularly important for upcoming measurements on $B^{3+}$.

The differential ms nuclear charge radius $\delta\langle r^2\rangle^{12,13}$ is determined from the isotope shift of the cg frequency of the entire fine structure using Eq. (2) and (7). The required NRQED atomic structure calculations of the mass-shift contribution $\delta\nu_M = 51\,719.29(25)$ MHz and the field-shift constant $F = -211.5(1)$ MHz/fm$^2$ were carried out by Yerokhin et al. up to the order of $m\alpha^6$[54]. The finite-nuclear-size effect in the isotope shift is the difference between our experimental value and the

calculated mass-shift contribution. It amounts to only 26.3(1.4) MHz, which is a $2 \times 10^{-8}$ fraction of the transition frequency. From the results described above, we derive a differential nuclear charge radius of

$$\delta\langle r^2\rangle^{12,13} = -0.1245(66)\,\text{fm}^2 \tag{3}$$

$$\delta R_c^{12,13} = -0.0253(14)\,\text{fm}. \tag{4}$$

Our results for the rms charge radii of $^{13}C$ and the change in rms charge radii are plotted in Fig. 3 and compared to results from electron scattering and muonic atom X-ray spectroscopy. The numerical values are included in Table 2. For $^{12}C$, the most accurate muonic measurement by Ruckstuhl et al.[26] disagrees with the weighted average of all electron-scattering results by about 2.4$\sigma$ of the combined uncertainty. Combining the e$^-$-scattering charge radius of $^{12}C$ with the $^{12,13}C^{4+}$ isotope-shift measurement provides a charge radius for $^{13}C$ that is in excellent agreement with the result of the $^{13}C$ e$^-$-scattering[55] but has 6 times reduced uncertainty, which is now similar to the uncertainty of the muonic atom result[56]. The discrepancy in terms of the combined uncertainty is even slightly larger than in $^{12}C$ (2.8$\sigma$ compared to 2.4$\sigma$). $\delta\langle r^2\rangle^{12,13}$ obtained from the difference in the muonic radii is smaller and has three times larger uncertainty than the laser-spectroscopic result, and just agrees within the combined uncertainty. Thus, we find a systematic offset in radii based on electromagnetic interaction with electrons versus those with muons. This is a different situation as in the $\alpha$-helion discrepancy, since the charge radius difference measured in ordinary ions agrees with that observed in muonic atoms, but the absolute charge radius from electron scattering does not. In the past, $^{12}C$ was always considered an excellent reference for charge radii measurements due to the high accuracy of the e$^-$-scattering result. This calls for verifying the e$^-$-scattering and the muonic atom results.

## Discussion

From a theoretical point of view, the carbon isotopes, especially $^{12}C$, due to their pronounced cluster structure, have long been theoretically interesting and challenging to describe via ab initio methods. Early explorations within the framework of fermionic molecular dynamics provided a quantitative description of various structures of $^{12}C$, including the Hoyle state, albeit with necessary phenomenological tuning of the interaction to a broad range of nuclear structure properties[57–59]. More recent lattice simulations based on nuclear interactions from chiral effective field theory (EFT) predicted the structure of the ground and Hoyle state of $^{12}C$ without adjustment based on the natural description of $\alpha$-clustering in such simulations[38,60,61]. Over the past 15 years, ab initio calculations of nuclei as heavy as $^{208}Pb$ have been performed using systematically improvable many-body methods[62–64] like the in-medium similarity renormalization group (IMSRG)[65]. Early IMSRG studies that included calculations of $^{12}C$ found atypically large differences in ground-state energies predicted by different approaches, indicating that the many-body description of $^{12}C$ is challenging[66–68]. New experimental measurements and theoretical predictions of $^{12}C$ and $^{13}C$ provide an interesting avenue to study not only the changing structure of carbon isotopes, but also how it emerges from nuclear forces and many-body methods.

We predict the properties of $^{12,13}C$ using ab initio valence-space IMSRG (VS-IMSRG)[67] and in-medium no-core shell-model (IM-NCSM)[68] nuclear structure calculations, which solve the many-body Schrödinger equation for a given input nuclear Hamiltonian in an approximate, but systematically improvable manner. We employ Hamiltonians with nucleon-nucleon and three-nucleon interactions from chiral EFT, using seven Hamiltonians that differ in their construction and how they are fitted to data to give insight into interaction uncertainties. VS-IMSRG(2) calculations were performed using 1.8/

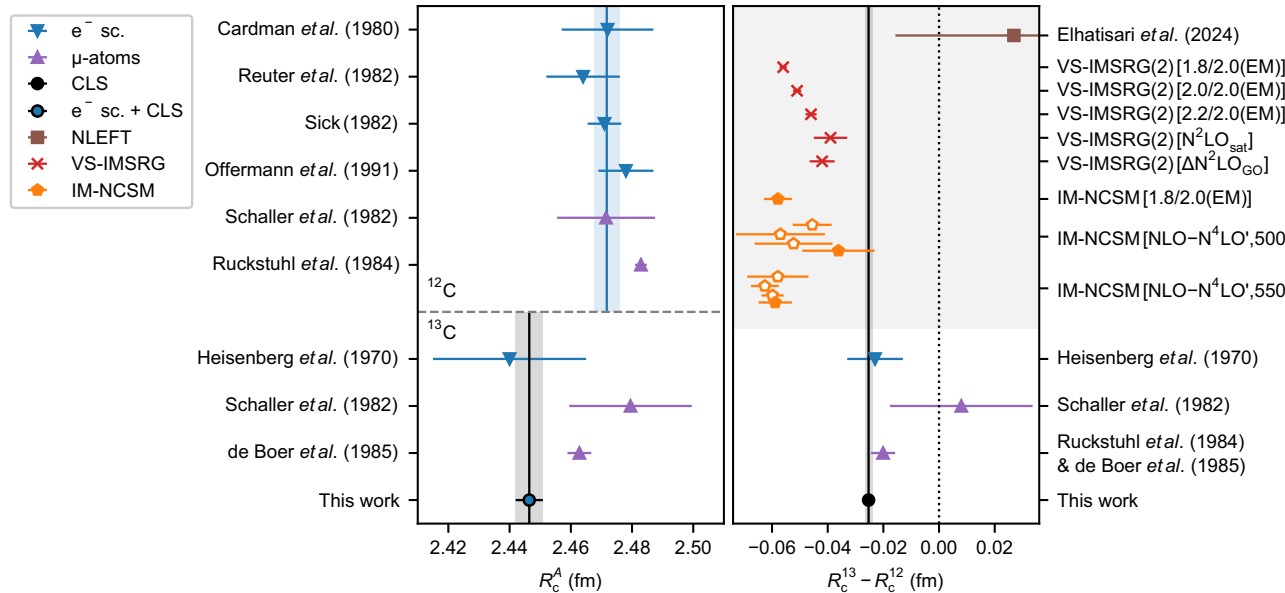

**Fig. 3 | Absolute and differential nuclear charge radii of $^{12,13}$C.** Experimentally determined absolute $R_c^{12}$, $R_c^{13}$ and nuclear charge radius difference $\delta R_c^{12,13} = R_c^{13} - R_c^{12}$ of $^{12,13}$C determined with elastic electron scattering (blue), muonic atom spectroscopy ($\mu$-atoms, purple) and collinear laser spectroscopy (CLS, black). Results from CLS and $e^-$-scattering were combined to obtain an improved $R_c^{13}$ (black & blue) purely from electronic measurements. The differential rms nuclear charge radii from $e^-$-scattering and $\mu$-atoms are differences of absolute radii while the CLS result

is determined directly from the isotope shift and ab initio atomic structure calculations using Eq. (2). $\delta R_c^{12,13}$ is also compared to ab initio valence-space in-medium similarity renormalization group (VS-IMSRG, red) and in-medium no-core shell-model (IM-NSCM, orange) calculations. The lower-order IM-NSCM results are plotted with open symbols. Results from nuclear-lattice effective field theory (NLEFT, brown) were published by Elhatisari et al.[61]. The numerical values of this plot are listed in Table 2.

## Table 2 | Absolute and differential nuclear charge radii of $^{12,13}$C determined with different ab initio nuclear structure calculations and experiments

| Method | $R_c^{12}$ | $R_c^{13}$ | $R_c^{13} - R_c^{12}$ |
|---|---|---|---|
| NLEFT [N$^3$LO][61] | 2.490(12) | 2.521(41) | 0.027(43) |
| VS-IMSRG(2) [1.8/2.0(EM)] | 2.416(1) | 2.361(1) | −0.056(1) |
| VS-IMSRG(2) [2.0/2.0(EM)] | 2.421(1) | 2.370(1) | −0.051(2) |
| VS-IMSRG(2) [2.2/2.0(EM)] | 2.423(1) | 2.377(1) | −0.046(2) |
| VS-IMSRG(2) [N$^2$LO$_{sat}$] | 2.405(1) | 2.366(6) | −0.039(6) |
| VS-IMSRG(2) [ΔN$^2$LO$_{GO}$] | 2.396(3) | 2.354(2) | −0.042(5) |
| IM-NCSM [1.8/2.0(EM)] | 2.421(16) | 2.363(12) | −0.058(5) |
| IM-NCSM [NLO, 500] | 2.348(249) | 2.302(248) | −0.046(7) |
| IM-NCSM [N$^2$LO, 500] | 2.521(103) | 2.464(101) | −0.057(16) |
| IM-NCSM [N$^3$LO, 500] | 2.532(38) | 2.479(27) | −0.052(14) |
| IM-NCSM [N$^4$LO′, 500] | 2.550(13) | 2.513(20) | −0.036(13) |
| IM-NCSM [NLO, 550] | 2.471(239) | 2.413(229) | −0.058(11) |
| IM-NCSM [N$^2$LO, 550] | 2.426(63) | 2.364(62) | −0.063(5) |
| IM-NCSM [N$^3$LO, 550] | 2.457(31) | 2.397(30) | −0.060(4) |
| IM-NCSM [N$^4$LO′, 550] | 2.482(23) | 2.423(20) | −0.059(6) |
| $e^-$-scattering[25,55,90–92] | 2.4717(42) | 2.440(25) | −0.023(10) |
| $\mu$-atom[93] | 2.472(16) | 2.480(20) | 0.008(26) |
| $\mu$-atom[26,56] | 2.4829(19) | 2.4628(39) | −0.0201(43) |
| CLS [this work] | — | 2.4464(45) | −0.0253(14) |

The maximum employed many-body orders or interactions of the theory results are given in square brackets. Radii are given in fm.

2.0(EM), 2.0/2.0(EM), 2.2/2.0(EM)[69], N$^2$LO$_{sat}$[70], and ΔN$^2$LO$_{GO}$[71], while for IM-NCSM calculations a family of non-local interactions up to N$^4$LO′ was employed[72]. We confirm the consistency of the two approaches by comparing VS-IMSRG(2) and IM-NCSM calculations with the 1.8/2.0(EM) Hamiltonian. More details on the Hamiltonians, methods and charge radii calculations are provided in Table 2 and Methods. Here, we concentrate on the results for the charge radii difference between the isotopes, which are shown in the right part of Fig. 3 and are compared with the experimental results. A plot of the absolute nuclear charge radii is shown in Fig. 4.

The experimentally observed size reduction of 0.0253(14) fm is overestimated by up to a factor of 2 in our calculations. The closest results are obtained with the VS-IMSRG using N$^2$LO$_{sat}$ and the IM-NCSM calculations with the N$^4$LO′(500) interactions. We note that the VS-IMSRG(2) calculations have uncertainties that are estimated solely from the model-space convergence. The uncertainties of the IM-NCSM calculations additionally include the Hamiltonian uncertainty and the convergence of the many-body expansion, with the exception of the 1.8/2.0(EM) Hamiltonian where only the many-body uncertainty is quantified. Thus, the scatter of the VS-IMSRG results beyond uncertainties is not unexpected. The VS-IMSRG(2) calculations underestimate the absolute charge radii for all Hamiltonians, suggesting a systematic underprediction due to the VS-IMSRG(2) approximation. VS-IMSRG(3) calculations were performed with the 1.8/2.0(EM) interaction to investigate the many-body uncertainty[73]. This brings the absolute radii in better agreement with the experiment, but further increases the discrepancy from the observed $\delta R_c^{12,13}$ by almost a factor of 2.

The IM-NCSM calculations in Table 2 show interesting dependencies of the charge radii, increasing with chiral order and decreasing with higher cutoff scale. For 550 MeV/$c$ we find good agreement with the muonic atom experiment for the charge radii of both isotopes at

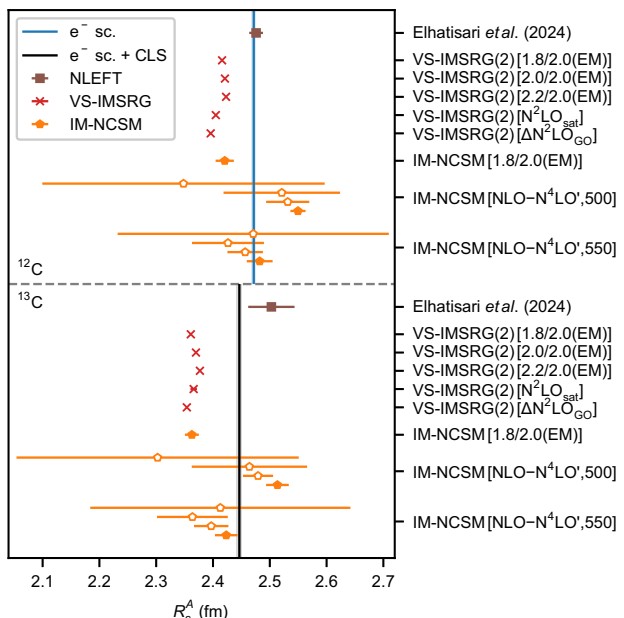

**Fig. 4 | Theoretical absolute nuclear charge radii of $^{12,13}$C.** The radii $R_c^{12,13}$ have been determined from ab initio valence-space in-medium similarity renormalization group (VS-IMSRG, red) and in-medium no-core shell model (IM-NSCM, orange) calculations. The lower-order IM-NSCM results are plotted with open symbols. Results from nuclear-lattice effective field theory (NLEFT, brown) were published by Elhatisari et al.[61]. The numerical values of this plot are listed in Table 2.

N$^4$LO′, but for the radius difference, the situation is reversed, providing a value compatible with experiment for cutoff 500 MeV/$c$. At N$^2$LO and N$^3$LO, however, both cutoffs provide a similar radius difference, pointing more towards the overestimated value.

We highlight that the theoretical prediction of the charge radius difference $\delta R_c^{12,13}$ benefits from the cancellation of correlated systematic uncertainties in the predictions of the absolute charge radii $R_c^{12}$ and $R_c^{13}$. This is clearly visible in the uncertainties of our IM-NCSM calculations, where Hamiltonian and many-body convergence uncertainties are quantified. To quantify these uncertainties for $\delta R_c^{12,13}$, we apply a Bayesian uncertainty quantification protocol (based on ref. 74 and described in Methods) directly to the charge radius difference, implicitly accounting for the cancellation of correlated uncertainties. As a result, the charge radius difference uncertainty is typically smaller than that of the absolute charge radii and also smaller than the resulting uncertainty if the uncertainties on $R_c^{12}$ and $R_c^{13}$ were uncorrelated.

All our ab initio calculations consistently predict a negative charge radius difference. The underprediction by both methods for all Hamiltonians, however, suggests that some relevant many-body correlations necessary for the precise prediction of this small difference are missing, requiring the development of improved many-body approximations to resolve.

The systematic discrepancy between the charge radii obtained in $^{12,13}$C using electron or muon interactions, uncovered by our measurements, must be consolidated or resolved by improved measurements on muonic atoms that are currently prepared by the QUARTET collaboration[24]. Laser spectroscopy of $^{14}$C is currently ongoing at COALA and will provide the nuclear charge radius of this isotope with comparable precision as obtained here for $^{13}$C and thus improve it by more than an order of magnitude compared to previous measurements.

## Methods

### Determination of the isotope shift

The absolute transition frequencies $\nu_i$ used in Eq. (7) are determined using frequency comb-referenced quasi-simultaneous collinear (c) and

anticollinear (a) laser spectroscopy. Fluorescence spectra are typically taken with 61 steps across a span of 800 MHz. All recorded spectra were fitted using the least-square algorithm `optimize.leastsq` provided by the `scipy` Python package[75]. A pure Gaussian model

$$g(\nu, \nu_{c/a}, \sigma, a, b) = a + b \exp\left[-\frac{(\nu - \nu_{c/a})^2}{2\sigma^2}\right] \quad (5)$$

was used to fit the data, since the natural linewidths of 9 MHz[51] of the transitions are negligible compared with the width of the velocity distribution of the ions. The typical full width at half maximum of a resonance is 150 MHz. The resonance centers $\nu_a$ and $\nu_c$ obtained from the corresponding anticollinear and collinear spectra, respectively, are combined to calculate the rest-frame frequency

$$\nu_0 = \sqrt{\nu_c \nu_a} - \frac{h\nu_c\nu_a}{2mc^2}. \quad (6)$$

The small second term, which amounts to 0.3 MHz for $^{13}$C$^{4+}$, takes care of the photon-recoil contribution that is transferred into kinetic energy of the ion during absorption. At least 28 anticollinear-collinear (ac) or collinear-anticollinear (ca) measurement pairs were taken for each hyperfine transition to minimize statistical fluctuations, and in all cases, the standard deviation of the mean was $\lesssim 1$ MHz.

There is some freedom in the choice of the isotope shift $\delta\nu^{A,A'}$ as it can be defined as any difference of linear combinations of transition frequencies sensitive to the nuclear charge radius. To get the smallest possible uncertainty, we calculate the weighted mean of all (hyper) fine-structure transition frequencies $\nu_i \in 1s2s\,^3S_1 \rightarrow 1s2p\,^3P_{0,1,2}$ for $^{12,13}$C. The weights are the theoretical transition strengths assuming no hyperfine-induced mixing. The cg frequency of the $^3S_1 \rightarrow {}^3P_J$ multiplet for each isotope is calculated as

$$\nu^A = \sum_i \nu_i \frac{(2F+1)(2F'+1)(2J'+1)}{3(2I+1)(2J+1)} \begin{Bmatrix} J' & J & 1 \\ F & F' & I \end{Bmatrix}^2, \quad (7)$$

where $J, J'$ and $F, F'$ are the electronic and total angular momentum quantum numbers of lower and upper state, respectively, and $I$ is the nuclear spin. This weighted mean is insensitive to hyperfine-induced mixing between the $1s2p\,^3P_{0,1,2}$ states and gives the same result as fitting the standard formula for HFS splittings to the resonance frequencies. In a second analysis, we have considered hyperfine-induced mixing explicitly, applying the theoretical magnetic dipole matrix elements $\langle \gamma'J'||T^{(1)}||\gamma''J''\rangle$ tabulated in ref. 52. By fitting

$$\nu_{(J,F)\rightarrow(\gamma\bar{J},F')} = \nu_{J\rightarrow(\gamma\bar{J},F')} - \frac{A_J}{2}[F(F+1) - J(J+1) - I(I+1)] \quad (8)$$

to the resonance frequencies, where $\nu_{J\rightarrow(\gamma\bar{J},F')}$ are the eigenvalues of the matrices with the entries[52]

$$\Omega^{F'}_{(\gamma'J')(\gamma''J'')} = \nu_{J\rightarrow(\gamma'J')}\delta_{(\gamma'J')(\gamma''J'')}$$
$$+ (-1)^{I+J'+F'} \begin{Bmatrix} I & J' & F' \\ J'' & I & 1 \end{Bmatrix} \sqrt{\frac{(2I+1)(I+1)}{I}} \mu_I \langle\gamma'J'||T^{(1)}||\gamma''J''\rangle, \quad (9)$$

hypothetical cg frequencies $\nu_{J\rightarrow(\gamma'J')}$ can be extracted that would result from Eq. (7) if there was no mixing between different $J'$ states. Here $\bar{J}$ is a label for the experimentally accessible mixed-$J$ states, $A_J$ is the HFS constant of the magnetic dipole contribution to the energy of the $^3S_1$ state, $\delta_{(\gamma'J')(\gamma''J'')}$ is the Kronecker delta, $\{:::\}$ is the Wigner-6j symbol, $\mu_I = 0.702\,369\,(4)\,\mu_N$[76] is the magnetic dipole moment of the nucleus and $\gamma' = 2S' + 1$ is the spin multiplicity of the state $^{\gamma'}P_{J'}$. In the fit of Eq. (8) to the resonance frequencies, $A_J$, $\nu_{J\rightarrow(\gamma'J')}$ and the diagonal elements

$\langle 3J'||T^{(1)}||3J'\rangle$ are taken as free parameters. Mixing with the $1s2p\,^1P_1$ state was taken into account using a fixed $\nu_{1\to11}$ and $\langle11||T^{(1)}||11\rangle$. This procedure relies on the calculated off-diagonal matrix elements $\langle\gamma J'||T^{(1)}||\gamma J''\rangle$, for which no uncertainty is specified in ref. 52. Assuming a relative uncertainty of $10^{-4}$ for the matrix elements of $T^{(1)}$, which are specified to five significant digits, yields shifts of the cg frequencies of $\lesssim 0.5\,$MHz, and are, thus, considerably smaller than the experimental uncertainty. The combined cg frequency of all $^3S_1 \to\,^3P_{0,1,2}$ transitions is additionally affected by mixing with the $1s2p\,^1P_1$ level. This contribution is expected to be small due to its large distance in energy from the $^3P$ levels. It can be estimated from the difference between including and excluding the $^1P_1$ state in the diagonalization and amounts to only 0.4 MHz, completely resolving the difference to the standard cg from Eq. (7). We note that the finite magnetic-moment distribution, included in the Zemach radius of a nucleus, will change the size of the hyperfine constant $A$ but does not affect the center of gravity of the transition.

## Systematic uncertainties

In our experiment, the ion velocity and, hence, the laser frequencies in the rest frame of the ions were scanned by changing the voltage potential in the fluorescence detection region (Doppler tuning). The frequencies of the lasers for collinear and anticollinear excitation were stabilized at frequencies that ensure resonant excitation at nearly the same ion velocity and, thus, the same Doppler tuning voltage. The uncertainties of the laser frequencies itself are determined from the statistics of the continuously measured beat signals of the frequency comb and directly considered in the Gaussian error propagation of Eq. (6). Systematic drifts of the laser frequencies are avoided by ensuring a sufficiently high beat signal used for stabilization to the atomic clock reference. Remaining differences $\delta U$ of the resonance positions were always below 0.5 V. The exact $\delta U$ is determined from the peak position parameter of the Gaussian lineshape model, which is fitted separately to both the collinear and anticollinear resonance signals. All other parameters of the lineshape model, such as the Gaussian linewidth $\sigma$, can also differ between collinear and anticollinear measurements, e.g., due to differences in frequency stability, laser power or laser-beam vibrations. The Gaussian width might differ, even for exact beam overlap, because of slightly different beam sizes and, therefore, additional velocity classes that are addressed by the larger beam. These parameters are, however, uncorrelated with the peak position parameter. To resolve both signals and to get the best signal-to-noise ratio, collinear and anticollinear measurements are recorded only quasi-simultaneously, meaning in quick succession. This has the disadvantage that a drifting high-voltage potential in the ion source is directly visible in $\delta U$. The main contribution to the drift is a changing electron-beam current in the EBIS. To compensate for this effect, measurements were always taken in the order ac-ca, which eliminates this error entirely as long as the voltage drift is linear. Additionally, the applied acceleration voltage was stabilized with a simple proportional regulator[77]. As a result, no systematic frequency shift due to uncompensated nonlinear voltage drifts was observed.

The largest systematic uncertainty originates from the imperfect alignment of the two laser beams. To make sure both laser beams are interacting with the same ion velocities, the profiles of the laser beams were adjusted to be roughly of the same size in the fluorescence detection region with a radius of 0.7 mm and superposed outside of the beamline at two points in the beam paths, 14 m apart from each other. The maximum displacement of 0.5 mm corresponds to an angle of $\lesssim 0.07$ mrad. During a single day of measurements, drifts of the laser-beam positions at the points of alignment, originating e.g., from angular drifts of the Nd:YVO$_4$ pump lasers or thermal drifts inside the Ti-sapphire cavity, remained below the estimated maximum displacement and were regularly checked and corrected once or twice per day. The ion beam was aligned with the collinear laser using a

**Table 3 | Summary of all systematic uncertainties of the transition frequency measurements in $^{13}C^{4+}$**

| Contribution | Symbol | Uncertainty (MHz) |
|---|---|---|
| Spatial velocity distribution | $\Delta\nu_{\text{spatial}}$ | 1.72 |
| Laser-/Ion-beam alignment | $\Delta\nu_{\text{angle}}$ | 0.09 |
| Photon recoils | $\Delta\nu_{\text{rec}}$ | 0.41 |
| Laser polarization | $\Delta\nu_{\text{pol}}$ | 0.24 |
| Absolute voltage | $\Delta\nu_U$ | 0.00 |
| Amplification factor | $\Delta\nu_{\delta U}$ | 0.00 |
| Total systematic uncertainty | $\Delta\nu$ | 1.79 |

The total systematic uncertainty is given by the geometric sum of all individual uncertainties.

combination of multi-channel plates (MCPs) and phosphor screens in the two beam diagnostic stations, which are 2.6 m apart. Here, a maximum misalignment of 0.62 mrad was estimated. While Doppler-induced frequency shifts due to the angular deviation are suppressed and amount to $\lesssim 0.1$ MHz, the concomitant spatial separation at the detection region can lead to larger effects due to the horizontal velocity dispersion in the ion beam caused by the 60°-bender at the entrance of the collinear beamline. The frequency shift associated with this effect is directly proportional to the horizontal positional difference of the two laser beams at the points of alignment, including a change in sign at perfect alignment, and was simulated and measured to appear as an additional statistical fluctuation centered around the atomic transition frequency with a standard deviation of 1.72 MHz. Due to the daily realignment of the ion and laser beams over the course of the three-month measurement period, the symmetric distribution of laser alignment configurations was adequately sampled. Additional uncertainties due to the Zeeman effect, photon recoils beyond the considered correction in Eq. (6) and the residual Doppler shift caused by the angular misalignment of the ion- and laser beams are listed in Table 3 and are added in square to yield a total systematic uncertainty of 1.8 MHz. For further details, see[47,48,78]. Due to the statistical nature of the systematic uncertainties, the total uncertainties of $\nu_i$ provided in Table 1 are the systematic and statistical uncertainties added in square.

## Nuclear Hamiltonians

We employ Hamiltonians with nucleon-nucleon and three-nucleon interactions from chiral effective field theory (EFT), where the full intrinsic Hamiltonian has the form $H = T - T_{\text{cm}} + V_{\text{NN}} + V_{\text{3N}}$ with the total kinetic energy $T$, the center-of-mass kinetic energy of the $A$-body nucleus $T_{\text{cm}}$, the nucleon-nucleon (NN) potential $V_{\text{NN}}$, and the three-nucleon (3N) potential $V_{\text{3N}}$. Chiral EFT Hamiltonians are truncated at a finite order in the EFT expansion, making nuclear Hamiltonians intrinsically uncertain. We employ several Hamiltonians that differ in their construction and fit to data to probe this uncertainty. The 1.8/2.0(EM), 2.0/2.0(EM), and 2.2/2.0(EM) Hamiltonians are constructed from the N³LO NN potential developed by Entem and Machleidt (EM) in ref. 79 unitarily transformed to the resolution scales $\lambda = 1.8$, 2.0, and 2.2 $\hbar$fm$^{-1}$, respectively, via the similarity renormalization group[80] and 3N potentials at N²LO with a regulator cutoff of 2.0 $\hbar$fm$^{-1}$ [69]. The NN potential is fitted to NN scattering data and the deuteron binding energy, and the 3N potentials are fitted to reproduce the binding energy of $^3H$ and the point-proton radius of $^4He$ for each of the transformed NN potentials. The names of these Hamiltonians are constructed from the NN resolution scale $\lambda$ and the 3N cutoff $\Lambda$, "$\lambda/\Lambda$(EM)", with the "(EM)" indicating the starting NN potential[79]. The N²LO$_{\text{sat}}$ Hamiltonian is constructed from N²LO NN and 3N potentials with a regulator cutoff of $\Lambda = 450$ MeV/$c$ and is fitted to NN scattering data, deuteron properties, ground-state energies and charge radii of few-body systems with $A \le 4$, and selected ground-state energies and charge radii for $^{14}C$ and $^{16,22,24,25}O$[70], where the fit to medium-mass nuclei helps to improve nuclear matter saturation properties of the

interaction. The $\Delta$N$^2$LO$_{GO}$ Hamiltonian, developed by the Gothenburg and Oak Ridge groups (GO), is constructed from N$^2$LO NN and 3N potentials with a cutoff of $\Lambda = 2.0\,\hbar$fm$^{-1}$ with the explicit inclusion of $\Delta$ isobars in the EFT[71]. It is fitted to NN scattering data, properties of few-body systems with $A \leq 4$, and nuclear matter properties, and additionally optimized to reproduce bulk properties of medium-mass nuclei. The family of non-local interactions up to N$^4$LO$'$[72] used in the IM-NCSM are constructed using the NN potentials from ref. 81 up to N$^4$LO and 3N interactions at N$^2$LO and N$^3$LO[82] with non-local regulators of $\Lambda = 500$ and 550 MeV/$c$, where N$^4$LO$'$ indicates a hybrid interaction with NN at N$^4$LO and 3N at N$^3$LO. The NN and 3N interactions are consistently unitarily transformed using the similarity renormalization group to a resolution scale of $\alpha = 0.08$ fm$^4$ (corresponding to $\lambda = 1.88\,\hbar$fm$^{-1}$). The NN interactions are fitted to NN scattering data and deuteron properties, and the 3N interactions are fitted to the ground-state energies of $^3$H and $^{16}$O.

## Nuclear structure calculations

Both the VS-IMSRG and the IM-NCSM are variants of the IMSRG[65,83], which produces a unitary transformation of the Hamiltonian to solve the Schrödinger equation. This transformation, generally parametrized as $U = \exp(\Omega)$, is normal ordered with respect to a reference state $|\Phi_0\rangle$, allowing it in practice to be truncated at the normal-ordered two-body level. All other operators, in particular charge radius operators, are consistently transformed using the same transformation, allowing for the computation of ground-state expectation values.

In this work, nuclear charge radii $R_c$ are determined with[84]

$$R_c = \sqrt{\langle R_p^2\rangle + \langle r_p^2\rangle + \frac{N}{Z}\langle r_n^2\rangle + \frac{3\hbar^2}{4m_p^2 c^2} + \langle r^2\rangle_{so}},\qquad(10)$$

with the point-proton squared charge radius $\langle R_p^2\rangle$, the proton and neutron squared charge radii $\langle r_p^2\rangle = (0.8409\,\text{fm})^2$ and $\langle r_n^2\rangle = -0.1155\,\text{fm}^2$[85], the relativistic Darwin-Foldy correction $3\hbar^2/4m_p^2 c^2 = 0.0332\,\text{fm}^2$, and the spin-orbit correction $\langle r^2\rangle_{so}$. Our IM-NCSM calculations neglect spin-orbit charge radius corrections, which contribute 0.0023–0.0033 fm$^2$ for $^{12}$C, 0.015–0.025 fm$^2$ for $^{13}$C, and 0.0027–0.0045 fm for $\delta R_c^{12,13}$ depending on the interaction, well within the assessed uncertainties. We note that in the optimization of the N$^2$LO$_{sat}$ Hamiltonian, a larger proton radius has been used, while here we employ the most recent value reported by the particle data group. This might lead to slightly smaller absolute radii, but will largely cancel in $\delta R_c^{12,13}$. All theoretical nuclear charge radii are compiled in Table 2. The differential and absolute radii are plotted in Figs. 3 and 4, respectively.

The VS-IMSRG[67,86] transforms the Hamiltonian such that a nucleus-specific valence-space Hamiltonian is decoupled, which can then be diagonalized using standard shell-model techniques. Our VS-IMSRG calculations start from a Hartree-Fock (HF) reference state. In particular, we employ Hartree-Fock single-particle states for states occupied in our reference state. For the remaining states, we employ the perturbatively improved natural orbital (NAT) basis[87], orthogonalizing the NAT basis with respect to the occupied HF states. We use the VS-IMSRG truncated at the normal-ordered two-body level, the VS-IMSRG(2), to decouple a $p$-shell valence space. A final diagonalization is performed with KSHELL[88]. Recent developments have made the VS-IMSRG calculations truncated at the normal-ordered three-body level available[73], and we explore the effect of the normal-ordered two-body truncation in our calculations by performing VS-IMSRG(3)-$N^7$ calculations. This captures induced three-body terms, leading to a more accurate solution of the many-body Schrödinger equation. For the final shell-model diagonalization, we truncate the residual three-body terms of all operators and perform the diagonalization with up to two-body operators.

The IM-NCSM[68] starts out with a standard NCSM calculation in a small reference space $N_{max} = 2$, using a natural orbital basis[89]. In a second step, this reference space is decoupled from higher-lying many-body states using the Magnus version of the multi-reference IMSRG truncated at the normal-ordered two-body level. The resulting Hamiltonian and consistently transformed operators are then used in a final NCSM calculation $N_{max} = 4$ to obtain the relevant observables. The uncertainties due to many-body truncations are probed by an explicit variation of the reference space and the final model-space truncation, as well as a variation of the IMSRG flow parameter. The chiral truncation uncertainties are extracted from the order-by-order variation of the observables through a Bayesian model[74]. For this, we use IM-NCSM calculations for all chiral orders starting at NLO for the aforementioned family of non-local interactions. Note that we apply this uncertainty quantification protocol directly to the radius differences as well, thus exploiting correlations of the radii in the two isotopes.

## Data availability
The data sets generated in the experiment and analyzed for the current study are publicly available at https://doi.org/10.48328/tudatalib-1500.

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

## Acknowledgements

We acknowledge support by the Deutsche Forschungsgemeinschaft (DFG, German Research Foundation)—Project-ID 279384907—SFB 1245. The experiment (P.I., K.K., B.M., P.M., W.N.) was also supported in part by the BMBF under Contract Nos. 05P19RDFN1 and 05P21RDFNA. P.M. and P.I. acknowledge support from HGS-HIRE. M.H., T.M., and A.S. were supported in part by the European Research Council (ERC) under the European Union's Horizon 2020 research and innovation program (Grant Agreement No. 101020842).

## Author contributions

The experiment was planned, prepared and performed by P.I., K.K., B.M., P.M., and W.N. The data analysis was performed by P.M. The theoretical calculations were developed and performed by M.H., T.M., R.R., and A.S. The paper was written by P.M., M.H., T.M., W.N., R.R., and A.S. All authors read and approved the manuscript.

## Funding

## Competing interests

The authors declare no competing interests.
