## [Transparent Peer Review file · Nature Communications]

The nuclear charge radius of ^{13}C

Corresponding Author: Dr Patrick Müller

Version 0:

Reviewer comments:

Reviewer #1

(Remarks to the Author)

This manuscript presents the first laser-spectroscopic measurement of the ^{13}C nuclear charge radius alongside a detailed ab initio theoretical analysis of the $^{12,13}\text{C}$ radius difference. The experimental result aligns well with an electron-scattering measurement from 1970 but with significantly reduced uncertainty, enabling the identification of a 3σ discrepancy relative to a muonic-atom measurement from 1985. The experimental method also opens pathways for isotope-shift studies of more exotic carbon isotopes. The combination of state-of-the-art experimental and theoretical approaches suggests missing many-body correlations in the theoretical description of these nuclei.

I consider the findings highly deserving of publication in Nature Communications. The manuscript is well-written and presents strong evidence for the claims of a systematic discrepancy between $^{12,13}\text{C}$ charge radii obtained using electron and muon interactions and ab initio theory.

I have three comments:

The use of several interactions and many-body methods strengthens this analysis. Perhaps I am missing something here, but why not compare IMNCSM and VSIMSRG with identical interactions to quantify part of the method uncertainty?

How is the correlation structure of the uncertainty in the theoretical result for $R_c^{12} - R_c^{13}$ analyzed? The covariance of the theoretical description of these two observables seems potentially important. Correlations and theory uncertainty is mentioned briefly at the end of the methods section with a reference to Melendez et al. I strongly recommend the authors to include a concise and quantitative discussion about the importance of these correlations, alongside information about the authors choice of prior and likelihood in their Bayesian model of the chiral truncation error.

Figure 3 is very good. However, could the authors include a similar plot comparing experimental and theoretical absolute nuclear charge radii for ^{12}C and ^{13}C as a complement to Table 2?

Reviewer #2

(Remarks to the Author)

This work intends to provide a highly accurate nuclear radius measurement of ^{13}C . It is based on high-precision measurements of 2s-2p transitions in He-like ^{13}C and ^{12}C ions by laser spectroscopy. It uses the fact that the $1s\ 2s\ 3S_1$ level in He-like ions is very long lived, as it can decay to the ground state only by a forbidden M1 transition. The laser spectroscopy is performed in an EBIT, where the ions are produced and trapped. The two-photon spectroscopy on trapped ions allows to obtain very accurate value, by a good control of the Doppler effect. This work provides the most accurate value ever obtained for the nuclear radius difference between ^{13}C and ^{12}C . It has a 3 sigma difference with the most accurate value available before, obtained with muonic atoms. The results point to possible systematic discrepancies between the muonic and electron scattering values. This paper thus provides important results, in particular in view of the new experimental projects for improving the accuracy of nuclear radii in light elements with muonic atoms, using new microcalorimeter detectors, which provide huge improvement in resolution compared to past experiments. This work also contains comparison with different nuclear models results. ^{12}C and ^{13}C are very important system to test advanced nuclear structure calculations.

I thus recommend publication of this work.

I have a specific question about the way the theory has been used to remove the hyperfine structure dependence. The theoretical values from Johnson et al in 1997 do not include the Bohr-Weiskopf effect for light elements (in fact only the case of gold has been evaluated). The Bohr-Weiskopf effect is due to the nuclear magnetic moment distribution. For ^{13}C it should be around 0.02% of the hyperfine shift of a given level, assuming a magnetic moment distribution with the same radius as the charge distribution. Have the authors looked into this effect and checked if this correction cancels out in a way compatible with the quoted accuracy, when evaluating the center of gravity of the HFS shifted values?

Reviewer #3

(Remarks to the Author)

Review Report on the Manuscript "The Nuclear Charge Radius of (^{13}C)"

The manuscript under review presents a study on the nuclear charge radius of ^{13}C through high-precision measurements of hyperfine transitions using laser excitation in helium-like ions of C^{4+} . The primary goal is to determine the rms value of the charge radius of the ^{13}C nucleus, whose motivation is very well stated. The study tackles an highly relevant topic in nuclear and atomic physics and showcases technically challenging experimental techniques with significant implications. The paper is well written and organized, with relevant information in the Methods section regarding the specifics of the experiment and data analysis.

However, there are issues with the clarity of nomenclature, assumptions in the data analysis, and the description of experimental procedures that should be addressed. Providing additional details and explanations on these points would enhance the manuscript's readability, transparency, and scientific rigor. I recommend revisions to address the following comments before considering the manuscript for publication.

General comments

- The authors mention (page 3, line 250, and in the Methods section) that the lasers require daily realignment. It is unclear in the manuscript why this realignment is necessary and whether the alignment shifts significantly over a single day of measurements. The authors should clarify if the potential drift in alignment is accounted for in the systematic uncertainty estimates and discuss its implications for the results, or if this alignment is done so that the quadrature sum of its uncertainty to the rest of the uncertainty budget makes statistical sense.
- The nomenclature of the nuclear Hamiltonians, such as "1.8/2.0 (EM)" (e.g., in the Methods section), is not adequately defined in the manuscript. Readers must refer to external sources, specifically references [71] and [75], to infer its meaning. It appears that "EM" refers to the work of Entem and Machleidt regarding the $N^3\text{LO}$ potential, but this should be explicitly stated. Providing a concise explanation of the notation and its origins within the text would greatly improve clarity and accessibility, especially for readers unfamiliar with the cited works.
- From page 8, line 515 onward, the manuscript discusses the determination of δU , obtained via the peak position parameter of a Gaussian line shape model fitted to collinear and anticollinear resonance signals. It is unclear whether the σ parameter of the Gaussian model is determined separately for each signal or if the fit is performed simultaneously across both datasets. If σ is determined separately, the authors should justify why this parameter might differ between the two datasets. If the fits are simultaneous, this should be explicitly stated, along with any assumptions about the consistency of σ across the datasets.

Specific Observations

- On page 7, line 451, the authors refer to "a-c measurement pairs," which presumably stands for "anticollinear-collinear." This abbreviation is not defined earlier in the manuscript. The authors should provide a clear definition when the term first appears to avoid confusion and ensure readability.
- The rms charge radius is referred several times in the manuscript although there is no prior definition (strangely, rms which is much more common is defined in the abstract).
- The frequency drift of the lasers doesn't seem to be included directly on the uncertainty budget (Table III). Is it included indirectly in some contribution?

Reviewer #4

(Remarks to the Author)

Version 1:

Reviewer comments:

Reviewer #1

(Remarks to the Author)

The authors have responded to all my questions and comments, and I recommend this manuscript for publication in Nature Communications.

Reviewer #2

(Remarks to the Author)

I think that the authors have clarified all the points listed by the referees. I think there paper can be published in Nature Communications in view of its importance.

Reviewer #3

(Remarks to the Author)

The authors have clarified all of my concerns and in my opinion the manuscript merits publication in Nat.Comms.

Reviewer #4

(Remarks to the Author)

Reviewer #1 (Remarks to the Author):

This manuscript presents the first laser-spectroscopic measurement of the ^{13}C nuclear charge radius alongside a detailed ab initio theoretical analysis of the $^{12,13}\text{C}$ radius difference. The experimental result aligns well with an electron-scattering measurement from 1970 but with significantly reduced uncertainty, enabling the identification of a 3σ discrepancy relative to a muonic-atom measurement from 1985. The experimental method also opens pathways for isotope-shift studies of more exotic carbon isotopes. The combination of state-of-the-art experimental and theoretical approaches suggests missing many-body correlations in the theoretical description of these nuclei.

I consider the findings highly deserving of publication in Nature Communications. The manuscript is well-written and presents strong evidence for the claims of a systematic discrepancy between $^{12,13}\text{C}$ charge radii obtained using electron and muon interactions and ab initio theory.

I have three comments:

- The use of several interactions and many-body methods strengthens this analysis. Perhaps I am missing something here, but why not compare IMNCSM and VSIMSRG with identical interactions to quantify part of the method uncertainty?

We have added values for IM-NCSM calculations with the 1.8/2.0 (EM) Hamiltonian to Fig. 3, Table II, and the added Fig. 4 with absolute charge radii. These are very consistent with the VS-IMSRG(2) results within the assessed many-body uncertainties. We also highlight this consistency in the main text:

“We confirm the consistency of the two approaches by comparing VS-IMSRG(2) and IM-NCSM calculations with the 1.8/2.0 (EM) Hamiltonian.”

- How is the correlation structure of the uncertainty in the theoretical result for $R_c^{12} - R_c^{13}$ analyzed? The covariance of the theoretical description of these two observables seems potentially important. Correlations and theory uncertainty is mentioned briefly at the end of the methods section with a reference to Melendez et al. I strongly recommend the authors to include a concise and quantitative discussion about the importance of these correlations, alongside information about the authors choice of prior and likelihood in their Bayesian model of the chiral truncation error.

The reviewer makes a very good point that the cancellation of correlated systematic uncertainties is essential for our predictions of the charge radius difference $R_c^{13} - R_c^{12}$. This correlation is not explicitly analyzed, but implicitly accounted for in our uncertainty quantification protocol where we estimate the uncertainties by directly considering the effective field

theory and many-body convergence behavior of the difference rather than the absolute radii separately. We have emphasized this important point in the main text:

“We highlight that the theoretical prediction of the charge radius difference $\delta R_c^{12,13}$ benefits from the cancellation of correlated systematic uncertainties in the predictions of the absolute charge radii R_c^{12} and R_c^{13} . This is clearly visible in the uncertainties of our IM-NCSM calculations, where Hamiltonian and many-body convergence uncertainties are quantified. To quantify these uncertainties for $\delta R_c^{12,13}$, we apply a Bayesian uncertainty quantification protocol (based on Ref. [76] and described in Methods) directly to the charge radius difference, implicitly accounting for the cancellation of correlated uncertainties. As a result, the charge radius difference uncertainty is typically smaller than that of the absolute charge radii and also smaller than the resulting uncertainty if the uncertainties on R_c^{12} and R_c^{13} were uncorrelated.”

- Figure 3 is very good. However, could the authors include a similar plot comparing experimental and theoretical absolute nuclear charge radii for 12C and 13C as a complement to Table 2?

As suggested, we added another plot showing the theoretical absolute nuclear charge radii and the two lines in Fig. 3 (left) as the experimental reference.

Reviewer #2 (Remarks to the Author):

This work intends to provide a highly accurate nuclear radius measurement of 13C. It is based on high-precision measurements of 2s-2p transitions in He-like 13C and 12C ions by laser spectroscopy. It uses the fact that the 1s 2s 3S1 level in He-like ions is very long lived, as it can decay to the ground state only by a forbidden M1 transition. The laser spectroscopy is performed in an EBIT, where the ions are produced and trapped. The two-photon spectroscopy on trapped ions allows to obtain very accurate value, by a good control of the Doppler effect. This work provides the most accurate value ever obtained for the nuclear radius difference between 13C and 12C. It has a 3 sigma difference with the most accurate value available before, obtained with muonic atoms. The results point to possible systematic discrepancies between the muonic and electron scattering values. This paper thus provides important results, in particular in view of the new experimental projects for improving the accuracy of nuclear radii in light elements with muonic atoms, using new microcalorimeter detectors, which provide huge improvement in resolution compared to past experiments. This work also contains comparison with different nuclear models results. 12C and 13C are very important system to test advanced nuclear structure calculations.

I thus recommend publication of this work.

- I have a specific question about the way the theory has been used to remove the hyperfine structure dependence. The theoretical values from

Johnson et al in 1997 do not include the Bohr-Weiskopf effect for light elements (in fact only the case of gold has been evaluated). The Bohr-Weiskopf effect is due to the nuclear magnetic moment distribution. For ^{13}C it should be around 0.02% of the hyperfine shift of a given level, assuming a magnetic moment distribution with the same radius as the charge distribution. Have the authors looked into this effect and checked if this correction cancels out in a way compatible with the quoted accuracy, when evaluating the center of gravity of the HFS shifted values?

The hyperfine anomaly changes the A factor of the hyperfine structure, but it does not shift the center of gravity. Therefore we do not expect any effect on it when empirically calculating the center-of-gravity of the fine-structure splitting using the experimental information from all individual hyperfine components. We also expect that the effect of a 0.02% change in the hyperfine factor A onto the mixing of the states is negligible as long as the levels with identical F values are much further apart than the size of this shift. Only if two levels are almost degenerate, such a small shift could have an influence, which is not the case in $^{13}\text{C}^{4+}$.

Theoretically, the distribution of the magnetic moment can be taken into account using the “Zemach radius”, which replaces a point magnetic dipole interaction with a distribution across the nucleus. The key point is that the Zemach radius term is proportional to the matrix elements of the I.S., and so it does not shift the center-of-gravity for the hyperfine structure. It just changes its magnitude relative to the rms nuclear charge radius. For the lighter He-like system Li^+ , this has been elaborated on in X.-Q. Qi et al., PRL 125, 183002 (2020) and the hyperfine structure splitting has been used to extract the Zemach radius. We have confirmed with Prof. Gordon Drake – one of the authors of the cited work – that our assumption is correct. Our measurements might initiate similar investigations for the nucleus of ^{13}C . We added a short note in the manuscript:

“We note that the finite magnetic-moment distribution, included in the Zemach radius of a nucleus, will change the size of the hyperfine constant A but does not affect the center of gravity of the transition.”

Reviewer #3 (Remarks to the Author):

Review Report on the Manuscript ”The Nuclear Charge Radius of (^{13}C)”

The manuscript under review presents a study on the nuclear charge radius of ^{13}C through high-precision measurements of hyperfine transitions using laser excitation in helium-like ions of C^{4+} . The primary goal is to determine the rms value of the charge radius of the ^{13}C nucleus, whose motivation is very well stated. The study tackles an highly relevant topic in nuclear and atomic physics and showcases technically challenging experimental techniques with significant implications. The paper is well written and organized, with relevant information

in the Methods section regarding the specifics of the experiment and data analysis. However, there are issues with the clarity of nomenclature, assumptions in the data analysis, and the description of experimental procedures that should be addressed. Providing additional details and explanations on these points would enhance the manuscript's readability, transparency, and scientific rigor. I recommend revisions to address the following comments before considering the manuscript for publication.

General comments

- The authors mention (page 3, line 250, and in the Methods section) that the lasers require daily realignment. It is unclear in the manuscript why this realignment is necessary and whether the alignment shifts significantly over a single day of measurements. The authors should clarify if the potential drift in alignment is accounted for in the systematic uncertainty estimates and discuss its implications for the results, or if this alignment is done so that the quadrature sum of its uncertainty to the rest of the uncertainty budget makes statistical sense.

We elaborated on the existing discussion, adding that the origin of the drifts are for example the Nd:YVO₄ pump lasers or thermal drifts inside the Ti-sapphire cavity. The drifts never exceeded the specified maximum misalignment (hence are included therein) and the laser alignment fluctuates statistically, which linearly translates to a frequency shift:

“During a single day of measurements, drifts of the laser beam positions at the points of alignment, originating e.g., from angular drifts of the Nd:YVO₄ pump lasers or thermal drifts inside the Ti sapphire cavity, remained below the estimated maximum displacement and were regularly checked and corrected once or twice per day. [... original text ...] The frequency shift associated with this effect is directly proportional to the horizontal positional difference of the two laser beams at the points of alignment, including a change in sign at perfect alignment, and was simulated and measured to appear as an additional statistical fluctuation centered around the atomic transition frequency with a standard deviation of 1.72 MHz. Due to the daily realignment of the ion and laser beams over the course of the three month measurement period, the symmetric distribution of laser alignment configurations was adequately sampled.”

- The nomenclature of the nuclear Hamiltonians, such as “1.8/2.0 (EM)” (e.g., in the Methods section), is not adequately defined in the manuscript. Readers must refer to external sources, specifically references [71] and [75], to infer its meaning. It appears that “EM” refers to the work of Entem and Machleidt regarding the N³LO potential, but this should be explicitly stated. Providing a concise explanation of the notation and its origins within the text would greatly improve clarity and accessibility, especially

for readers unfamiliar with the cited works.

We have added statements to the discussion of nuclear Hamiltonians in the Methods section to clarify the nomenclature (additions underlined):

“The 1.8/2.0 (EM), 2.0/2.0 (EM), and 2.2/2.0 (EM) Hamiltonians are constructed from the $N^3\text{LO}$ NN potential developed by Entem and Machleidt (EM) in Ref. [81] unitarily transformed to the resolution scales $\lambda = 1.8, 2.0, \text{ and } 2.2 \text{ fm}^{-1}$, [...]. The names of these Hamiltonians are constructed from the NN resolution scale λ and the 3N cutoff Λ , “ λ/Λ (EM),” with the “(EM)” indicating the starting NN potential [81].”

“The $N^2\text{LO}_{\text{sat}}$ Hamiltonian is constructed from $N^2\text{LO}$ NN and 3N potentials with a regulator cutoff of $\Lambda = 450 \text{ MeV}/c$ and is fit to NN scattering data, deuteron properties, ground-state energies and charge radii of few-body systems with $A \leq 4$, and selected ground-state energies and charge radii for ^{14}C and $^{16,22,24,25}\text{O}$ [72], where the fit to medium-mass nuclei helps to improve nuclear matter saturation properties of the interaction.”

“The $\Delta N^2\text{LO}_{\text{GO}}$ Hamiltonian, developed by the Gothenburg and Oak Ridge groups (GO), is constructed from $N^2\text{LO}$ NN and 3N potentials with a cutoff of $\Lambda = 2.0 \text{ fm}^{-1}$ with the explicit inclusion of Δ isobars in the EFT [73].”

- From page 8, line 515 onward, the manuscript discusses the determination of δU , obtained via the peak position parameter of a Gaussian line shape model fitted to collinear and anticollinear resonance signals. It is unclear whether the σ parameter of the Gaussian model is determined separately for each signal or if the fit is performed simultaneously across both datasets. If σ is determined separately, the authors should justify why this parameter might differ between the two datasets. If the fits are simultaneous, this should be explicitly stated, along with any assumptions about the consistency of σ across the datasets.

We stated explicitly that fits are performed separately for each resonance signal. We also clarified that all parameters, including σ , can differ between collinear and anticollinear measurements due to differences in frequency stability, laser power or laser-beam vibrations, but are uncorrelated to the peak position parameter:

“All other parameters of the lineshape model, such as the Gaussian linewidth σ , can also differ between collinear and anticollinear measurements, e.g., due to differences in frequency stability, laser power or laser-beam vibrations. The Gaussian width might differ, even for exact beam overlap, because of slightly different beam sizes and therefore additional velocity classes that are addressed by the larger beam. These parameters are, however, uncorrelated to the peak position parameter.”

Specific Observations

- On page 7, line 451, the authors refer to "a-c measurement pairs," which presumably stands for "anticollinear-collinear." This abbreviation is not defined earlier in the manuscript. The authors should provide a clear definition when the term first appears to avoid confusion and ensure readability.

We specifically wrote "anticollinear-collinear (ac) or collinear-anticollinear (ca) measurement pairs" here, which also should make the later use of ac-ca clearer.

- The ms charge radius is referred several times in the manuscript although there is no prior definition (strangely, rms which is much more common is defined in the abstract).

The abbreviation ms is defined as mean-square upon its first appearance in the introduction.

- The frequency drift of the lasers doesn't seem to be included directly on the uncertainty budget (Table III). Is it included indirectly in some contribution?

We added two sentences regarding the uncertainties and drifts of the laser frequencies. The uncertainties are directly considered in Eq. (6) using Gaussian error propagation. Frequency drifts are negligible due to the stabilization to the atomic clock referenced frequency comb by ensuring a sufficiently high beat signal.

"The uncertainties of the laser frequencies itself are determined from the statistics of the continuously measured beat signals of the frequency comb and directly considered in the Gaussian error propagation of Eq. (6). Systematic drifts of the laser frequencies are avoided by ensuring a sufficiently high beat signal used for stabilization to the atomic clock reference."

Reviewer #4 (Remarks to the Author):

We hope that we have clarified all questions of the referees.